# High-throughput 96-well plate-based porcine antibody isolation protocol

**John Byrne**[1], Christina Bourne[1], Sitka Eguiluz[1], Stephanie N. Langel[2], Elisa Crisci[1]*

**1** Department of Population Health and Pathobiology, College of Veterinary Medicine, North Carolina State University, Raleigh, North Carolina, United States of America, **2** Department of Pathology, Center for Global Health and Diseases, Case Western Reserve University School of Medicine, Cleveland, Ohio, United States of America

* ecrisci@ncsu.edu

## Abstract

For decades, scientists have used column chromatography to purify an array of analytes. The same chromatography system has also been deployed for the isolation and purification of antibodies to increase sensitivity and specificity of detection assays such as western blot, enzyme-linked immunosorbent assay (ELISA), and immunohistochemistry. However, even with the combination of these modalities the detection of specific antibodies developed in response to treatment, like vaccination, remains difficult due to physiological differences among species, sample types evaluated, and differing physiological states. Therefore, we developed a high-throughput antibody isolation protocol to measure influenza-specific antibodies in pregnant and lactating pigs, because the samples, in particular serum, milk and colostrum, contain components that cause high background. We developed a high-throughput 96-well plate-based method using a modified column chromatography technique to specifically isolate swine immunoglobulin (Ig) isotypes. This method utilizes isotype-specific reagents to isolate the IgG and IgA antibody isotypes from various fluids collected at multiple time points from individual animals following immunization. After sample processing and antibody isolation, the results showed a rapid and consistent yield of specific IgG and IgA, with comparable outcomes between single-column chromatography and our 96-well plate system, the latter offering a time-saving advantage. Specifically, the standard single-column chromatography required 2 to 3 hours to isolate 12 samples, whereas our method enabled the isolation of 192 samples in just 8 to 9 hours making this the ideal method for an immunogenicity study utilizing a variety of animal samples from multiple timepoints.

## Introduction

New methods are constantly in development to better analyze biological systems, functionalities, and responses. We developed a high-throughput 96-well plate-based method using a modified column chromatography technique to specifically isolate swine immunoglobulin (Ig) isotypes. This method utilizes isotype-specific reagents to isolate IgG and IgA antibodies from various fluids collected at multiple time points from individual animals. Pigs are frequently used as a translational biomedical model for studying human anatomy, immunity, and disease due to the anatomical and physiological similarities to humans [1]. While many antibody-based immunoassays in pigs are run on whole samples, such as milk or serum [2,3],

**Funding:** This work was supported by the Bill and Melinda Gates Foundation (https://www.gatesfoundation.org/; Grant number: INV-22595D awarded to Stephanie Langel). The funders had no role in the study design, data collection and analysis, decision to publish, or preparation of the manuscript.

**Competing interests:** The authors have declared that no competing interests exist.

to assess maternal immunity, these samples may yield false-positive results due to high background. Therefore, we have developed the method, described here, for isolating Igs from these porcine fluids.

To test this method and validate this protocol, we used pigs (Table 1) to evaluate mucosal immunization with a proprietary, monovalent, influenza A virus (IAV) vaccine (Fig 1) to induce specific immunity in pregnant and lactating pigs after a prime-boost regimen, with the booster administered 28 days after the primary dose (day 0), one week post partum. The antigenicity of IAV is characterized by two variable and synergistic surface glycoproteins, hemagglutinin (HA) and neuraminidase (NA). The HA glycoprotein mediates virus binding to sialic acid receptors located on the cell surface [4]. Therefore, most vaccine strategies target HA using strain-specific recombinant HA to produce seasonal vaccines based on the annual monitoring of influenza virus strains circulating in the population [5].

To assess the antibody response induced by the vaccine, serum, colostrum (collected during or within 1-12 hours after parturition), and milk (collected weekly after parturition) samples were collected to verify the induction of HA-specific antibodies via ELISA (Fig 2).

Additionally, we standardized a resin-based approach for pig antibodies in our method, as many commercial antibody-binding reagents have not been assessed for pig antibody isolation. The initial use of standard individual column chromatography to isolate antibodies from a single pig at a single timepoint proved labor- and time-intensive, and therefore not suitable for the substantial number of samples required in a vaccination study. We have developed a new high-throughput 96-well plate-based method using resins capable of successfully isolating swine IgG and IgA from up to 192 samples at one time.

## Materials and methods

The protocol described in this peer-reviewed article is published on protocols.io, DOI: dx.doi.org/10.17504/protocols.io.yxmvmee6ng3p/v1 and is included for printing as supporting information file 1 with this article.

**Table 1. Description of the control and vaccinated groups.**

| Group | Pig Breed | Gestation in Days | N | Vaccination |
|---|---|---|---|---|
| Control | Yorkshire X | 113-115 | 4 | N/A |
| Vaccinated | Yorkshire X | 113-115 | 4 | Prime (Day 0) three weeks pre partum/ Boost (Day 28) one week post partum |

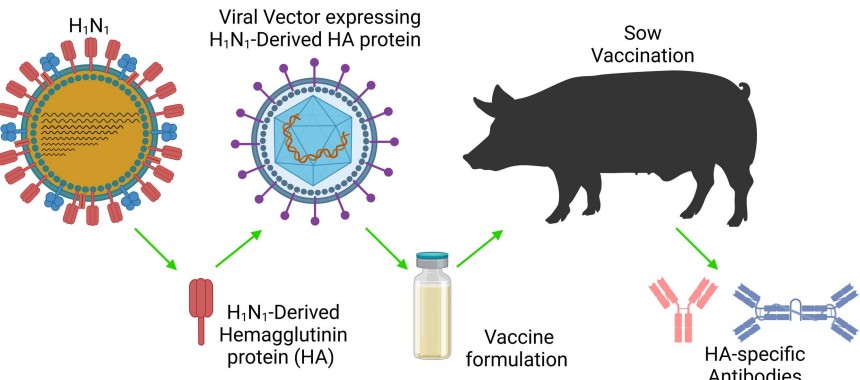

**Fig 1. The depiction of the experimental design and use of an Influenza A hemagglutinin (HA) monovalent vectored vaccine in the porcine animal model vaccinated then boosted 28 days later.** HA-specific IgG and IgA antibodies were evaluated in serum, milk and colostrum (Figure created with BioRender.com. Byrne, J. (2025) https://biorender.com/g36e786).

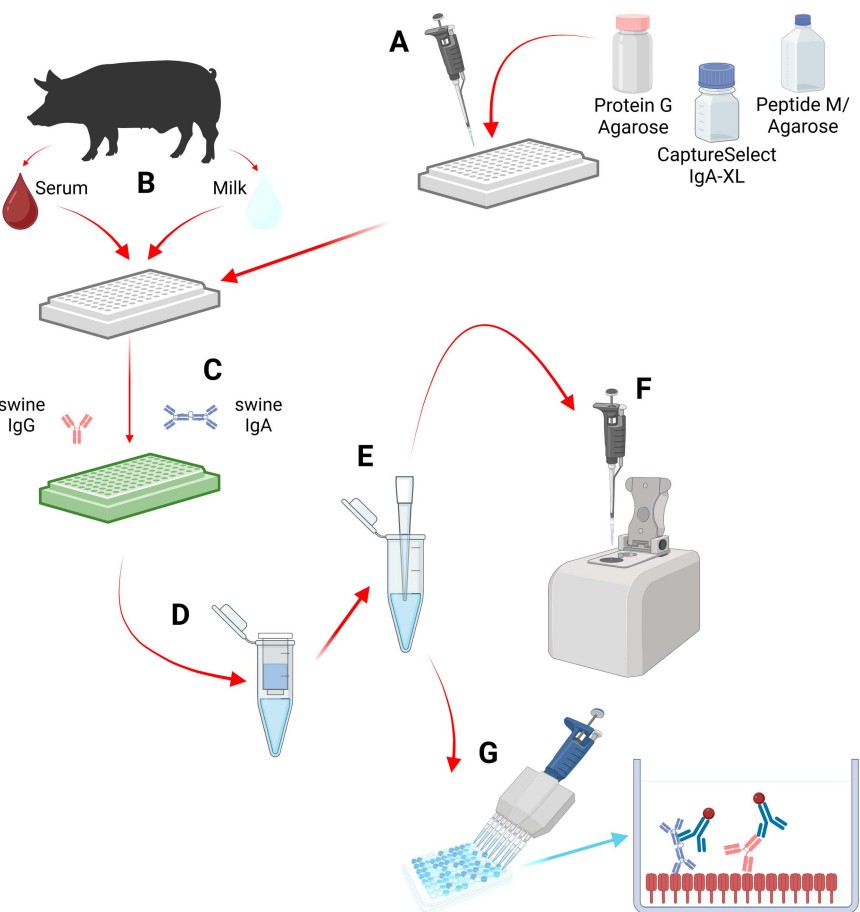

**Fig 2. The project design of the antibody isolation protocol leading to the evaluation of HA-specific antibody response post-vaccination via in-house HA ELISA. A.** Selecting a resin dependent on desired antibody isotype to isolate. **B.** Pig-derived sample collection to be diluted, plated, and incubated with the resin to bind the antibody. **C.** Centrifugation of the stacked plates with the elution buffer and collection of the isolated antibodies in the bottom collection plate. **D.** The eluted products are moved to individual, clean filter tubes for concentration and changing the storage buffer at the same time. **E.** The eluted antibody is moved and stored in a clean collection microcentrifuge tube. **F.** Each isolated antibody sample is pipetted onto the nanodrop spectrophotometer to record the published concentration in mg/mL. **G.** The isolated antibody is diluted to a specified concentration and plated in a precoated low-bind ELISA plate for optical density (OD) analysis, confirming vaccine-induced antibody specificity. (Figure created with BioRender.com. Byrne, J. (2025) https://biorender.com/s11c541).

- The animals used in this project have been approved under the North Carolina State University's Institutional Animal Care and Use Committee number 22-122 and number 20-473.

- Antibody detection was performed via in-house HA-specific ELISA. U-bottom 96-well Microtiter™ Microplates (Thermo scientific) were coated with 2 µg/well of purified HA-trimer (eENZYME) in 1x KPL Coating Solution (Fisher) and incubated at room temperature for 2 hours. Plates were washed 3 times with 0.1% PBST after each incubation. Protein coated plates were blocked with 100 µL blocking buffer (1% fish gelatin in 0.1% PBS Tween 20) and incubated at room temperature for 1 hour. To evaluate immunoglobulins in serum, milk and colostrum, samples were diluted 1:100 in blocking buffer, added to the plate, and incubated at room temperature for 1 hour. When evaluating HA-specific IgG and IgA, isolated antibody isotypes from the respective sample types were diluted to 5 µg/100 µL in

blocking buffer. Antibody binding was detected by adding 100 µL of goat anti-porcine IgG (Abcam) or IgA (Bethyl) specific HRP-conjugated antibodies diluted 1 : 4000 in blocking buffer at room temperature for 1 hour. The colorimetric detection was evaluated using 100 µl of KPL SureBlue Reserve™ TMB Microwell Substrate (VWR) per well. The analysis was performed at the end of exposure (30-minute incubation) the reaction was stopped by addition of 100 µL of KPL SureBlue Reserve™ STOP (VWR) requiring optical density (OD) evaluation utilizing the Biotek Microplate spectrophotometer set at 450 nm.

- Unpaired t test was performed using GraphPad Prism version 10.2.3 for Windows, GraphPad Software, Boston, Massachusetts USA, www.graphpad.com

**Equipment:**

- Sorvall Legend Micro 21R Microcentrifuge (Thermo Scientific)
- Sorvall Legend XTR (Refrigerated), 120V (Thermo Scientific)
- Thermo Scientific NanoDrop 2000c Spectrophotometer
- Biotek Synergy 2 SL Microplate Reader

**List Materials:**

- Pierce Protein G Agarose Cat# 20399 (Thermo Scientific)
- Peptide M/ Agarose Cat# gel-pdm-5 (InvivoGen)
- CaptureSelect IgA-XL Affinity Matrix Cat# 2943972010 (Thermo Scientific)
- Goat Anti-Pig IgG H&L (HRP) Cat# ab6915 (Abcam)
- Goat anti-Pig IgA Heavy Chain Antibody HRP Conjugated Cat# A100-102P (Bethyl)
- Purified Influenza Hemagglutinin trimer Cat# AA 18-530 (eENZYME)
- Glycine, 0.2M buffer solution., pH 2.5 Cat# J61855.AP (Thermo Scientific)
- Tris Hydrochloride, 1M Solution (pH 8.0/Mol. Biol.) Cat# BP1758-500 (Fisher Scientific)
- Tris-EDTA, 1x Solution, pH 8.0 ± 0.1 Cat# 77-86-1 (Fisher Scientific)
- Sodium chloride, ACS, 99.0% min Cat# 7647-14-5 (Fisher Scientific)
- Ethanol, Absolute (200 Proof), Molecular Biology Grade Cat# BP2818500 (Fisher Scientific)
- Ultra-pure water - In House
- Seracare Life Sciences Inc KPL coating solution concentrate Cat# 50-674-41 (Fisher Scientific)
- Corning Mediatech Cell Culture Phosphate Buffered Saline (10X) Cat# MT-46013 CM (Fisher Scientific)
- Tween 20™, Ultrapure, Thermo Scientific Chemicals Cat# AAJ20605AP (Fisher Scientific)
- Thermo Scientific™ Fish Serum Blocking Buffer Cat# PI37527 (Fisher Scientific)
- SureBlue Reserve™ Microwell Substrate, KPL Cat# 95059-294 (VWR)
- TMB Stop Solution 450 nm, KPL Cat# 95059-200 (VWR)
- Universal 200 ul Pipette tips Cat# 76322-144 (VWR)
- BrandTech Scientific 781722, 96-Well Plates, immunoGrade, Clear Transparent, Flat Bottom Cat# 91-415F (Genesee Scientific)

- Multiscreen 96 well Plate, hydrophobic PVDF membrane Cat# MSIPS4W10 (Millipore Sigma)
- Thermo Scientific™ 96-Well Microtiter™ Microplates Cat# 14-245-71 (Fisher Scientific)

## Results

All resins demonstrated reactivity with the pig antibodies. These resins are not produced specifically for isolating porcine antibodies. Protein G has been published by the manufacturer

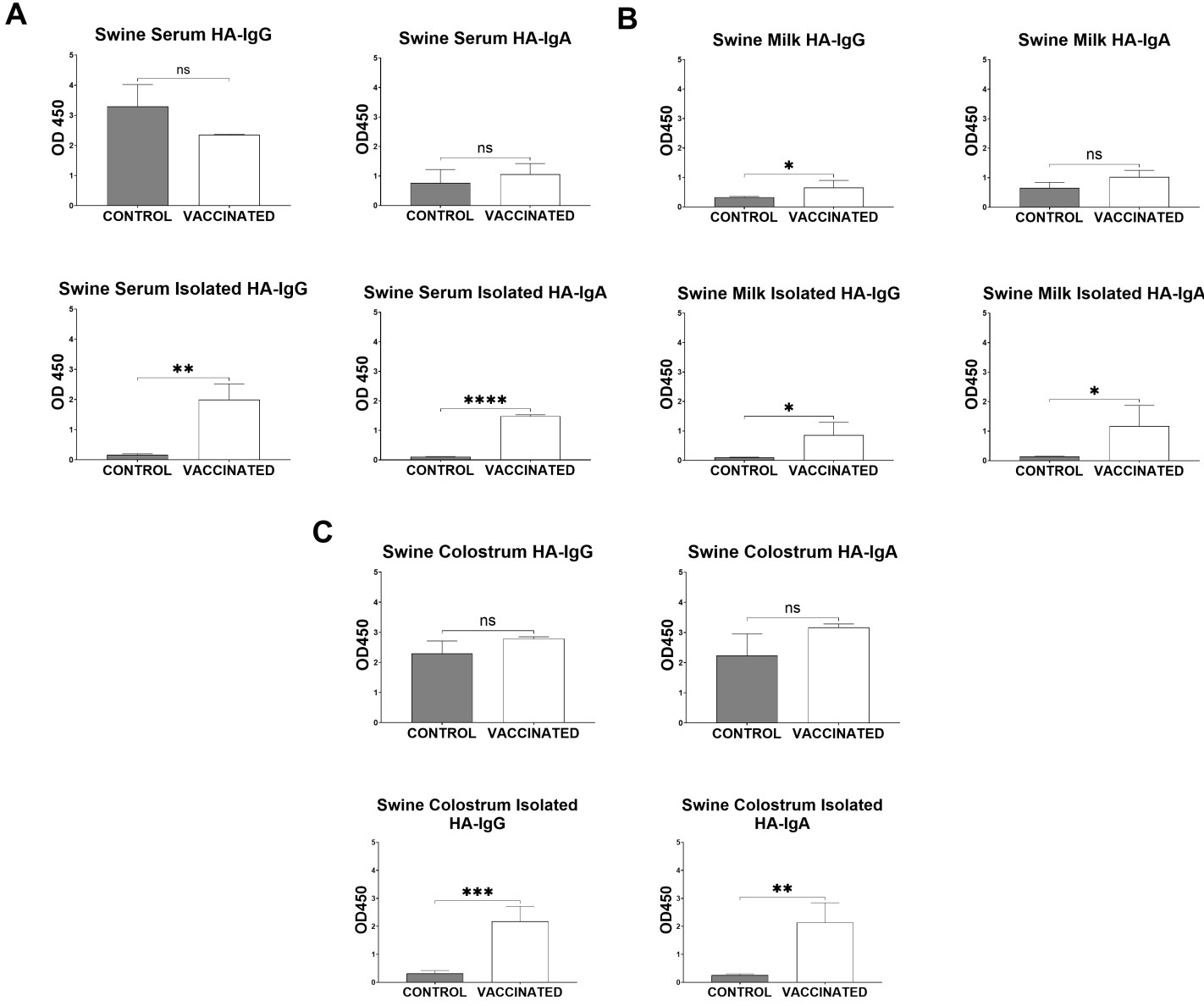

**Fig 3. Isolation of swine immunoglobulins from serum, colostrum, and milk significantly reduced background OD450 values. A.** ELISA results of vaccinated and control groups comparing whole serum, 2 weeks post-boost (1:100 dilution), to antibodies (50 µg/mL) isolated from serum using Protein G for IgG and Capture-Select for IgA **B.** ELISA results of vaccinated and control groups comparing whole milk, 2 weeks post-boost (1:100 dilution), to antibodies (50 µg/mL) isolated from milk using Protein G for IgG and CaptureSelect for IgA **C.** ELISA results of vaccinated and control groups comparing whole colostrum, 2 weeks post-prime (1:100 dilution), to antibodies (50 µg/mL) isolated from colostrum using Protein G for IgG and CaptureSelect for IgA. OD 450nm. Mean ± SD. N = 4. Statistics: Unpaired t test. * p < 0.05 **p < 0.01 ***p < 0.001 **** $p$ < 0.0001.

for polyclonal IgG purification from mouse, human, cow, goat, and sheep serum [6]. CaptureSelect has been published by the manufacturer for specificity of human IgA, IgA1, IgA2, dimeric IgA, and secretory IgA [7]. Peptide M has been published by the manufacturer for isolation of human IgA1 and IgA2 and bovine IgA [8]. Nevertheless, using these resins, we have successfully isolated pig Igs from vaccinated and unvaccinated control animals. Serum and milk were both collected 2 weeks post-boost vaccination while colostrum was collected during parturition, 2 weeks post-prime vaccination. All samples were plated at either a 1:100 dilution for "whole samples" or 50 ug/ml concentration for isolated Ig. Initial ELISA data using whole samples showed no difference in HA-specific antibody levels between control or vaccinated animal except for a significant difference in milk HA-specific IgG. Post-isolated ELISA data showed that HA-vaccinated animals produced significant HA-specific IgG and IgA antibody responses in serum, colostrum, and milk (Fig 3). Therefore, isolating the antibodies from serum, colostrum, and milk provided greater consistency, and reduced false-positive in the analysis of HA-specific antibodies. The Ig concentrations varied from 0.1 to 16 mg/mL for total IgG and 0.2 to 2.0 mg/mL for total IgA. The varied range in pig antibody concentrations were likely due to the sample type (i.e., milk, serum, colostrum), pregnancy and lactation time point, and inherent differences between animals. While standard single column chromatography can achieve the same outcome, this high-throughput method enables simultaneous antibody isolation from up to 384 (the capacity of four 96-well plates, matching the maximum centrifuge capacity) individual samples.

Finally, after Ig isolation, the ELISA data (Fig 3) shows detectable vaccine-induced HA-specific antibodies in samples from vaccinated pigs and lower HA-binding antibodies in samples from control animals.

## Supporting information

**S1 File. High throughput 96 well plate based porcine antibody isolation protocol.**
(PDF)

## Acknowledgments

We would like to acknowledge Dr. Susan Tonkonogy and Abigail Williams for their help in editing the manuscript.

## Author contributions

**Conceptualization:** John Byrne.

**Data curation:** John Byrne, Christina Bourne.

**Formal analysis:** John Byrne, Christina Bourne.

**Funding acquisition:** Stephanie N. Langel, Elisa Crisci.

**Investigation:** John Byrne, Christina Bourne.

**Methodology:** John Byrne, Christina Bourne, Sitka Eguiluz.

**Project administration:** John Byrne.

**Resources:** Elisa Crisci.

**Supervision:** Elisa Crisci.

**Validation:** John Byrne.

**Visualization:** John Byrne, Christina Bourne, Sitka Eguiluz, Elisa Crisci.

**Writing – original draft:** John Byrne, Christina Bourne, Sitka Eguiluz, Elisa Crisci.

**Writing – review & editing:** John Byrne, Christina Bourne, Sitka Eguiluz, Stephanie N. Langel, Elisa Crisci.

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
