## [Decision Letter · Decision Letter 0]

17 Dec 2024

PONE-D-24-53200High-throughput 96-well plate-based porcine antibody isolation protocolPLOS ONE

Dear Dr. Crisci,

Thank you for submitting your manuscript to PLOS ONE. After careful consideration, we feel that it has merit but does not fully meet PLOS ONE’s publication criteria as it currently stands. Therefore, we invite you to submit a revised version of the manuscript that addresses the points raised during the review process.

During the revision process, please address the comments related to description or experimental data, evaluation of results, and overall presentation of the manuscript.

We look forward to receiving your revised manuscript.

Kind regards,

Victor C Huber

Academic Editor

PLOS ONE

2.When completing the data availability statement of the submission form, you indicated that you will make your data available on acceptance. We strongly recommend all authors decide on a data sharing plan before acceptance, as the process can be lengthy and hold up publication timelines. Please note that, though access restrictions are acceptable now, your entire data will need to be made freely accessible if your manuscript is accepted for publication. This policy applies to all data except where public deposition would breach compliance with the protocol approved by your research ethics board. If you are unable to adhere to our open data policy, please kindly revise your statement to explain your reasoning and we will seek the editor's input on an exemption. Please be assured that, once you have provided your new statement, the assessment of your exemption will not hold up the peer review process.

4. We note you have not yet provided a protocols.io PDF version of your protocol and/or a protocols.io DOI. When you submit your revision, please provide a PDF version of your protocol as generated by protocols.io (the file will have the protocols.io logo in the upper right corner of the first page) as a Supporting Information file. The filename should be S1_file.pdf, and you should enter “S1 File” into the Description field. Any additional protocols should be numbered S2, S3, and so on. Please also follow the instructions for Supporting Information captions [https://journals.plos.org/plosone/s/supporting-information#loc-captions]. The title in the caption should read: “Step-by-step protocol, also available on protocols.io.”

Please assign your protocol a protocols.io DOI, if you have not already done so, and include the following line in the Materials and Methods section of your manuscript: “The protocol described in this peer-reviewed article is published on protocols.io (https://dx.doi.org/10.17504/protocols.io.[...]) and is included for printing purposes as S1 File.” You should also supply the DOI in the Protocols.io DOI field of the submission form when you submit your revision.

If you have not yet uploaded your protocol to protocols.io, you are invited to use the platform’s protocol entry service [https://www.protocols.io/we-enter-protocols] for doing so, at no charge. Through this service, the team at protocols.io will enter your protocol for you and format it in a way that takes advantage of the platform’s features. When submitting your protocol to the protocol entry service please include the customer code PLOS2022 in the Note field and indicate that your protocol is associated with a PLOS ONE Lab Protocol Submission. You should also include the title and manuscript number of your PLOS ONE submission.

Reviewers' comments:

Reviewer's Responses to Questions

**Comments to the Author**

1. Does the manuscript report a protocol which is of utility to the research community and adds value to the published literature?

Reviewer #1: Yes

Reviewer #2: Yes

2. Has the protocol been described in sufficient detail?

To answer this question, please click the link to protocols.io in the Materials and Methods section of the manuscript (if a link has been provided) or consult the step-by-step protocol in the Supporting Information files.

The step-by-step protocol should contain sufficient detail for another researcher to be able to reproduce all experiments and analyses.

Reviewer #1: Partly

Reviewer #2: Yes

3. Does the protocol describe a validated method?

Reviewer #1: No

Reviewer #2: Yes

4. If the manuscript contains new data, have the authors made this data fully available?

Reviewer #1: No

Reviewer #2: Yes

**5. Is the article presented in an intelligible fashion and written in standard English?**

Reviewer #1: **No: ** The author needs to be improved. Grammar and typos need to be corrected.

Reviewer #2: Yes

6. Review Comments to the Author

Reviewer #1: In this work, Byrne et al., describe a protocol to isolate porcine specific IgG and IgA from serum, colostrum and milk in a high-throughput manner. The authors employ a 96-well plate setup in which they incubate samples with Protein G, CaptureSelect IgA, or Peptide M, followed by antibody isolation, concentration, and quantification. Overall, the manuscript is interesting, but several points need to be considered prior to publication.

The language of the manuscript needs to be carefully revised, since there are many grammatical errors and typos all over the work.

Ideally, the protocol included as supplementary material should have been integrated in the manuscript prior to submission to be able to appreciate how the whole manuscript looks.

Table 1 does not seem necessary, since this information is already mentioned in the text.

Figure 1. The influenza virus particle does not display the internal ribonucleoproteins.

Figure 2, ideally should depict a more detailed (but straightforward) scheme showing all major the steps of the protocol, including sample and resin plate prep, antibody isolation and concentration, and quantification.

It seems that the authors take a single point dilution to express antibody titers. This is usually inaccurate since results are highly dependent of the dilution used for calculating the optical density. An endpoint titer or area under the curve (AUC) value obtained from serial dilutions of the samples should be presented for the ELISA results.

It is unclear how serum HA-specific IgG can be so high (even higher) in the controls vs vaccinated animals. Likewise, how colostrum IgG and IgA are so high. Many studies using complete serum in different animal models show that responses in serum are usually specific, and show a clear distinction between controls and vaccinated animals.

The protein used in the methods is not described in the protocol not the ELISA method used for assessing the antibody levels. This information is important.

We recommend to remove filler words such as ‘clearly’, ‘dramatically’ (L151-153) throughout the manuscript.

Reviewer #2: In this article, Byrne et al. present a protocol that they established for high-throughput purification/isolation of immunoglobulins from porcine fluids that are prone to non-specific or high background binding to antigen, making these fluids unreliable for direct testing. The authors validated their method against influenza hemagglutinin.

This article merits publication because it is applicable towards other species (for which chromatography and Ig-specific resin or reagents are available); the authors validated their method for isolation of IgA after intranasal immunization, with this isotype predominant in mucosal immunity, protective against certain pathogens, and maternally transferred; they demonstrate efficacy of an intranasal vaccine targeting influenza hemagglutinin; they demonstrate that the species cross-reactivity of Ig-binding resin such as protein G extends to porcine antibodies.

I have some minor comments that may help the authors potentially improve the article. Apart from one overarching comment, the rest of the comments are organized into the sections as they appear in the text.

OVERALL

I feel that there are two topics that are equally important: the protocol itself for Ig isolation and immunization against influenza. However, it is unclear if the authors chose to prioritize one or the other or both. On the one hand, this is a “Lab Protocol” type article, with influenza HA presented as an antigen that helps validate the method (although demonstrating isolation of total Ig may have been sufficient). I see this reflected in the title and the abstract. On the other hand, the introduction leads with influenza, front and center, rather than opting for the structure of the abstract. Figure 3 also presents results on vaccine efficacy which give the impression of a “Research Article” type article. I think the article is publishable and acceptable as either.

The authors can choose to reinforce one aspect or the other as they see fit, to draw in a specific readership. I see advantages to both because a more general article would attract a broader readership because the method is applicable to other species and antigens. A more specialized article could have a larger impact within the authors’ direct field of study. However, the latter would require more details on the immunization, ELISA, and perhaps additional data analysis (see comments that follow).

INTRODUCTION

Line 63-64 – May be worthwhile to edit this sentence because we do not know what fraction of the antigen-specific antibodies are transferred passively to piglets as opposed to lost, and we also do not know if the antibodies are protective, only that they are antigen-specific. Neither transfer nor protection are directly studied here.

MATERIAL AND METHODS

I think details are missing, some important and some less important, about how the ELISA was performed. Since the introduction gives the impression that influenza is a major component of this article and this is backed up by the ELISA data, it would help to provide more details on the assay, if possible. It probably does not fit into the existing protocols.io page, but it would be helpful to document it in the main article.

I could be mistaken but the RESULTS section describe when the fluids were collected relative to parturition but not when they were collected relative to booster immunization. This is a factor for antibody titer and kinetics of development of an adaptive immune response.

The concentrations/dilutions of fluids used in the ELISA fit better here than solely in the Figure 3 legend.

Were the ELISAs all performed on the same plate/performed roughly at the same time/developed for the same duration (i.e., the time between substrate addition and stopping of the reaction)? This is related to the comment in the RESULTS section. If all specimens were tested on the same plate at the same time and developed for the same duration (roughly), then it makes it possible to judge the relative level of background binding from different fluids, the relative yield of antigen-specific Igs, compare them statistically, etc. I.e., multiple graphs in Figure 3 could be combined and additional statistical comparisons made. This information would be invaluable to steer the article toward more a "Research Article" type.

Protocols.io page – in the “Quantify Antibody concentration” section, was the “Sample Type” in the program set to “IgG” for example? It may help readers get a more accurate estimate of the protein concentration by instructing the program to use an extinction coefficient that is closer to that of polyclonal antibodies than the default extinction coefficient/sample type.

RESULTS

Whether from their own data or from the literature, could the authors comment on whether the protein G/peptide M/CaptureSelect resin have different affinities to different Ig subclasses (e.g., IgG1 vs. IgG2, etc.)? This is why it could be worthwhile to add more details about how the ELISA was performed (see comment in the MATERIALS AND METHODS section), in the event the secondary reagents were subclass-specific for example. Could help peers in the field.

Could the authors comment more and specify the differences in Ig yield instead of simply ranges in concentration? This could be an additional table. From the three fluids/compartments tested? Other comparisons: homeostatic levels of different Ig isotypes, any changes post-vaccination, compare the compartments, etc. This could help other researchers compare their yields with yours, gauge the binding capacity of the resins, etc.

Related to the comment above, for the data in Figure 3, since one of the objectives stated in the abstract and introduction was to remove background binding, is it possible to perform additional analyses such as comparing the OD between different fluids to gauge the level of background binding between fluids/compartments? An ANOVA would then be the appropriate test.

FIGURES

Figure 2 – This figure could benefit from more labels like in Figure 1. For example, the tube and pipette tip in the center is ambiguous. Otherwise, more details can be added in the legend.

MISCELLANEOUS

Please, correct some very minor grammatical and typographical errors below:

Line 30 – In “…to measure influenza-specific antibodies specific to pregnant and lactating pigs…”, the word “specific” appears twice in succession but with different meanings. Maybe replacing the “specific to” with “from immunized” could provide better context and clarity for the reader.

Line 57 – Please, delete the “an” before “influenza”.

Line 82 – Please, delete the “the” before “prior”. Replace “isolated” with “isolation”.

Figure 2 legend – two hyphens are missing for “HA specific” and “post vaccination”.

Figure 1 – Hyphen missing before “Derived” and another one missing before “specific”.

Step 35 of the protocols.io page – Correct the misspelled “yeild”.

7. PLOS authors have the option to publish the peer review history of their article (what does this mean? ). If published, this will include your full peer review and any attached files.

**Do you want your identity to be public for this peer review?** For information about this choice, including consent withdrawal, please see our Privacy Policy .

Reviewer #1: No

Reviewer #2: **Yes: ** Justin Tze Ho Chan

---

## [Author Response · Author response to Decision Letter 1]

29 Jan 2025

Reviewer #1: In this work, Byrne et al., describe a protocol to isolate porcine specific IgG and IgA from serum, colostrum and milk in a high-throughput manner. The authors employ a 96-well plate setup in which they incubate samples with Protein G, CaptureSelect IgA, or Peptide M, followed by antibody isolation, concentration, and quantification. Overall, the manuscript is interesting, but several points need to be considered prior to publication.

1. The language of the manuscript needs to be carefully revised, since there are many grammatical errors and typos all over the work.

- The spelling and grammar have been reviewed and adjusted.

2. Ideally, the protocol included as supplementary material should have been integrated in the manuscript prior to submission to be able to appreciate how the whole manuscript looks.

- The supplemental data (protocol.io and figures) was submitted to the journal as specified by the submission guidelines and followed the instructions from the lab protocol template.

3. Table 1 does not seem necessary, since this information is already mentioned in the text.

- While the statement of the Table 1 inclusion is valid, we think that including the table allows for a comprehensive overview of the animals included that was not mentioned within the body of the manuscript.

4. Figure 1. The influenza virus particle does not display the internal ribonucleoproteins.

- The RNA has been added to the influenza depiction to more accurately display the viral particle. This figure will be located on line 85 in the final publication.

5. Figure 2, ideally should depict a more detailed (but straightforward) scheme showing all major the steps of the protocol, including sample and resin plate prep, antibody isolation and concentration, and quantification.

- Figure 2 has been updated with the suggested edits. The description of each stage depicted in figure 2 has been added to the figure legend.

6. It seems that the authors take a single point dilution to express antibody titers. This is usually inaccurate since results are highly dependent of the dilution used for calculating the optical density. An endpoint titer or area under the curve (AUC) value obtained from serial dilutions of the samples should be presented for the ELISA results.

- Thank you for your feedback regarding the use of a single-point dilution to express antibody titers. The primary focus of this paper is to demonstrate a high-throughput method for the isolation of swine antibodies, rather than to compare antibody immune responses between groups. While endpoint titers or area under the curve (AUC) values derived from serial dilutions are indeed more precise for detailed comparative analyses, they are beyond the scope of this study. Our goal was to showcase the efficiency of our approach in antibody isolation, and the single-point dilution serves as a representative metric for this purpose

7. It is unclear how serum HA-specific IgG can be so high (even higher) in the controls vs vaccinated animals. Likewise, how colostrum IgG and IgA are so high. Many studies using complete serum in different animal models show that responses in serum are usually specific, and show a clear distinction between controls and vaccinated animals.

- We appreciate this comment and concern as this is the basis of the manuscript and the reason behind the development and use of this protocol. As written in the abstract, “detection of specific antibodies developed in response to treatment, like a monovalent vaccination, remains difficult due to physiological differences among species, sample types evaluated, and differing physiological states. Therefore, we developed a high-throughput antibody isolation protocol to measure influenza-specific antibodies specific to pregnant and lactating pigs, because the samples, in particular serum, milk and colostrum, contain components that cause high background.” While speculations can be made, more research needs to be done in the physiology of pregnant and lactating pigs to determine the cause of this background.

8. The protein used in the methods is not described in the protocol not the ELISA method used for assessing the antibody levels. This information is important.

- The proteins used in the method are described in step 6 of the protocol published in protocols.io. A description of the ELISA method has been added to the method portion of the manuscript with the inclusion of the purified hemagglutinin protein used. Lines 104-119

9. We recommend to remove filler words such as ‘clearly’, ‘dramatically’ (L151-153) throughout the manuscript.

- These filler words have been reduced.

Reviewer #2: In this article, Byrne et al. present a protocol that they established for high-throughput purification/isolation of immunoglobulins from porcine fluids that are prone to non-specific or high background binding to antigen, making these fluids unreliable for direct testing. The authors validated their method against influenza hemagglutinin.

This article merits publication because it is applicable towards other species (for which chromatography and Ig-specific resin or reagents are available); the authors validated their method for isolation of IgA after intranasal immunization, with this isotype predominant in mucosal immunity, protective against certain pathogens, and maternally transferred; they demonstrate efficacy of an intranasal vaccine targeting influenza hemagglutinin; they demonstrate that the species cross-reactivity of Ig-binding resin such as protein G extends to porcine antibodies.

I have some minor comments that may help the authors potentially improve the article. Apart from one overarching comment, the rest of the comments are organized into the sections as they appear in the text.

1. I feel that there are two topics that are equally important: the protocol itself for Ig isolation and immunization against influenza. However, it is unclear if the authors chose to prioritize one or the other or both. On the one hand, this is a “Lab Protocol” type article, with influenza HA presented as an antigen that helps validate the method (although demonstrating isolation of total Ig may have been sufficient). I see this reflected in the title and the abstract. On the other hand, the introduction leads with influenza, front and center, rather than opting for the structure of the abstract. Figure 3 also presents results on vaccine efficacy which give the impression of a “Research Article” type article. I think the article is publishable and acceptable as either.

The authors can choose to reinforce one aspect or the other as they see fit, to draw in a specific readership. I see advantages to both because a more general article would attract a broader readership because the method is applicable to other species and antigens. A more specialized article could have a larger impact within the authors’ direct field of study. However, the latter would require more details on the immunization, ELISA, and perhaps additional data analysis (see comments that follow).

- We thank the reviewer for the useful comment. We decided to redirect the introduction to prioritize and reflect the “Lab Protocol” article style.

INTRODUCTION

2. Line 63-64 – May be worthwhile to edit this sentence because we do not know what fraction of the antigen-specific antibodies are transferred passively to piglets as opposed to lost, and we also do not know if the antibodies are protective, only that they are antigen-specific. Neither transfer nor protection are directly studied here.

- Line 63-64 of the original draft have been edited accordingly to more accurately reflect the data. The new sentence is included starting in line 77 of the revised draft.

MATERIAL AND METHODS

3. I think details are missing, some important and some less important, about how the ELISA was performed. Since the introduction gives the impression that influenza is a major component of this article and this is backed up by the ELISA data, it would help to provide more details on the assay, if possible. It probably does not fit into the existing protocols.io page, but it would be helpful to document it in the main article.

- The ELISA information has been added into the methods section of the article. Lines 104-119

4. I could be mistaken but the RESULTS section describe when the fluids were collected relative to parturition but not when they were collected relative to booster immunization. This is a factor for antibody titer and kinetics of development of an adaptive immune response.

- The timeline in which the fluids were collected has been added to the Figure 3 legend and results section.

5. The concentrations/dilutions of fluids used in the ELISA fit better here than solely in the Figure 3 legend.

- Concentrations and dilution information has been added to the results section. Lines 164-170

6. Were the ELISAs all performed on the same plate/performed roughly at the same time/developed for the same duration (i.e., the time between substrate addition and stopping of the reaction)? This is related to the comment in the RESULTS section. If all specimens were tested on the same plate at the same time and developed for the same duration (roughly), then it makes it possible to judge the relative level of background binding from different fluids, the relative yield of antigen-specific Igs, compare them statistically, etc. I.e., multiple graphs in Figure 3 could be combined and additional statistical comparisons made. This information would be invaluable to steer the article toward more a "Research Article" type.

- ELISAs were performed on different plates dependent on the isolated antibody or whole sample input. The authors agree that this manuscript will be best suitable for a protocol style manuscript rather than research article

7. Protocols.io page – in the “Quantify Antibody concentration” section, was the “Sample Type” in the program set to “IgG” for example? It may help readers get a more accurate estimate of the protein concentration by instructing the program to use an extinction coefficient that is closer to that of polyclonal antibodies than the default extinction coefficient/sample type.

- The correction was made on Protocols.io for the quantification as the nanodrop should be set to Protein A280. While the IgG setting may be more accurate considering the addition of the extinction coefficient in the calculation, the authors have chosen not to utilize this program. When normalizing total protein concentration for the ELISA adding the extinction coefficient may not be the best practice when comparing different isotypes. In the Isolation procedure and the analysis of antibody specificity, while IgG can be quantified on the nanodrop, the individual IgA oligomers cannot be assumed and accurately quantified utilizing a blanket extinction coefficient.

RESULTS

8. Whether from their own data or from the literature, could the authors comment on whether the protein G/peptide M/CaptureSelect resin have different affinities to different Ig subclasses (e.g., IgG1 vs. IgG2, etc.)? This is why it could be worthwhile to add more details about how the ELISA was performed (see comment in the MATERIALS AND METHODS section), in the event the secondary reagents were subclass-specific for example. Could help peers in the field.

- To our knowledge, there is no data on the binding strength of individual pig IgG subclasses to Protein G or other Immunoglobulin-binding resins currently in the market.

9. Could the authors comment more and specify the differences in Ig yield instead of simply ranges in concentration? This could be an additional table. From the three fluids/compartments tested? Other comparisons: homeostatic levels of different Ig isotypes, any changes post-vaccination, compare the compartments, etc. This could help other researchers compare their yields with yours, gauge the binding capacity of the resins, etc.

- The authors agree that the range would be the most appropriate way to define the expected yield. Since the variability in yield between isolations (even from the same sample) is so high, there would not be a more accurate way to show an expected result.

10. Related to the comment above, for the data in Figure 3, since one of the objectives stated in the abstract and introduction was to remove background binding, is it possible to perform additional analyses such as comparing the OD between different fluids to gauge the level of background binding between fluids/compartments? An ANOVA would then be the appropriate test.

- The authors agree to omit this type of information. Comparison through the different fluids derived from the variable tissues may not be a beneficial analysis without an understanding as to why we see this background result. There may be too much speculation as to justify a conclusion from this analysis.

FIGURES

11. Figure 2 – This figure could benefit from more labels like in Figure 1. For example, the tube and pipette tip in the center is ambiguous. Otherwise, more details can be added in the legend

- Labels and details have been added to Fig. 2. Lines 78-88

MISCELLANEOUS

12. Please, correct some very minor grammatical and typographical errors below:

Line 30 – In “…to measure influenza-specific antibodies specific to pregnant and lactating pigs…”, the word “specific” appears twice in succession but with different meanings. Maybe replacing the “specific to” with “from immunized” could provide better context and clarity for the reader.

Line 57 – Please, delete the “an” before “influenza”.

Line 82 – Please, delete the “the” before “prior”. Replace “isolated” with “isolation”.

Figure 2 legend – two hyphens are missing for “HA specific” and “post vaccination”.

Figure 1 – Hyphen missing before “Derived” and another one missing before “specific”.

Step 35 of the protocols.io page – Correct the misspelled “yeild”.

- All corrections have been reviewed and made throughout the manuscript and protocol as suggested by the reviewer.

---

## [Decision Letter · Decision Letter 1]

20 Feb 2025

High-throughput 96-well plate-based porcine antibody isolation protocol

PONE-D-24-53200R1

Dear Dr. Crisci,

We’re pleased to inform you that your manuscript has been judged scientifically suitable for publication and will be formally accepted for publication once it meets all outstanding technical requirements.

Kind regards,

Victor C Huber

Academic Editor

PLOS ONE

Additional Editor Comments (optional):

Reviewers' comments:

Reviewer's Responses to Questions

**Comments to the Author**

1. Does the manuscript report a protocol which is of utility to the research community and adds value to the published literature?

Reviewer #2: Yes

2. Has the protocol been described in sufficient detail?

To answer this question, please click the link to protocols.io in the Materials and Methods section of the manuscript (if a link has been provided) or consult the step-by-step protocol in the Supporting Information files.

The step-by-step protocol should contain sufficient detail for another researcher to be able to reproduce all experiments and analyses.

Reviewer #2: Yes

3. Does the protocol describe a validated method?

Reviewer #2: Yes

4. If the manuscript contains new data, have the authors made this data fully available?

Reviewer #2: Yes

**5. Is the article presented in an intelligible fashion and written in standard English?**

Reviewer #2: Yes

6. Review Comments to the Author

Reviewer #2: I am satisfied with how Byrne et al. responded to and addressed my suggestions and comments on the original draft of their manuscript. I have only minor comments and corrections to share with the authors for the revised manuscript. They are organized in the order that they appear in the text or in categories. However, even if this feedback is not 100% addressed, I would recommend/accept the manuscript for publication. Apologies for not catching or suggesting some of them earlier as I understand it may be too late for certain changes.

General

Since this is a “Lab Protocol” type article, involving a lot of methodology, it is natural to write in the passive voice, but perhaps alternating it with the active voice would make the manuscript more stimulating and increase readability in certain sections.

Introduction

Figure 1 legend – Since the figure may not be self-explanatory nor self-explanatory to everyone, readers may benefit from a more detailed legend, consistent with the Figure 2 legend. Every figure is the “the depiction” of something so perhaps the legend could be more informative and “standalone”, without readers having to pick up details from the main text.

Materials and Methods

I could be wrong, but bullet points are not a requirement for PLOS ONE for the methodology. Here, they are a bit odd and I am not sure if they are beneficial over paragraphs that are the format of organization everywhere else. However, I also have no way of seeing how bullet points would end up in the final typeset article so maybe it would not be so odd after all.

As a “Lab Protocol” type article, the materials are especially important and maybe the authors could consider integrating and organizing them into a categorized table, like in the Cell Press STAR methods “key resource table” format: https://www.cell.com/pb-assets/journals/research/cell/methods/table-template1-1699013648137.docx

The “List Materials” are especially random and could benefit from some organization and categorization.

Results

Line 164 – The “1:100 dilution” might contradict the “1:100 and 1:1000” on line 110.

Grammar

Line 64 – I am not familiar with what an “individual dose” refers to. Maybe a more fitting term would be “primary dose”, especially since the authors use it on line 163 as well.

Figure 2 legend – There are two hyphens missing: one on line 80 for “Pig-derived” and one on line 88 for “vaccine-induced”.

Line 95 – Replace “one” with “a”.

Line 163 – Add missing hyphens in “post-boost” and “post-prime”.

Figure 3 legend – On lines 179, 181 and 183, add the missing hyphens after “post”

Line 186 – Add the missing hyphen in “vaccine-induced”.

Protocols.io – Make title consistent with the one in the main text. Replace “insure” with “ensure” in Step 1, just like how the authors used “ensuring” in Step 5 and “ensure” in Step 23.

Thank you. I look forward to reading the final published work.

7. PLOS authors have the option to publish the peer review history of their article (what does this mean? ). If published, this will include your full peer review and any attached files.

**Do you want your identity to be public for this peer review?** For information about this choice, including consent withdrawal, please see our Privacy Policy .

Reviewer #2: **Yes: ** Justin Tze Ho Chan

---

## [Editor Report · Acceptance letter]

PONE-D-24-53200R1

PLOS ONE

Dear Dr. Crisci,

I'm pleased to inform you that your manuscript has been deemed suitable for publication in PLOS ONE. Congratulations! Your manuscript is now being handed over to our production team.

Kind regards,

on behalf of

Dr. Victor C Huber

Academic Editor

PLOS ONE